# hLMSC Secretome Affects Macrophage Activity Differentially Depending on Lung-Mimetic Environments

**DOI:** 10.3390/cells11121866

**Published:** 2022-06-08

**Authors:** Bryan Falcones, Zackarias Söderlund, Arturo Ibáñez-Fonseca, Isaac Almendros, Jordi Otero, Ramon Farré, Sara Rolandsson Enes, Linda Elowsson Rendin, Gunilla Westergren-Thorsson

**Affiliations:** 1Lung Biology, Biomedical Center, Department of Medical Science, Lund University, 22184 Lund, Sweden; zackarias.soderlund@med.lu.se (Z.S.); arturo.ibanez-fonseca@med.lu.se (A.I.-F.); sara.rolandsson_enes@med.lu.se (S.R.E.); linda.elowsson@med.lu.se (L.E.R.); gunilla.westergren-thorsson@med.lu.se (G.W.-T.); 2Unit of Biophysics and Bioengineering, Facultat de Medicina i Ciències de la Salud, Universitat de Barcelona, 08036 Barcelona, Spain; isaac.almendros@ub.edu (I.A.); jordi.otero@gmail.com (J.O.); rfarre@ub.edu (R.F.)

**Keywords:** MSC-based therapy, lung physiomimetic culture, preconditioning, macrophages

## Abstract

Mesenchymal stromal cell (MSC)-based therapies for inflammatory diseases rely mainly on the paracrine ability to modulate the activity of macrophages. Despite recent advances, there is scarce information regarding changes of the secretome content attributed to physiomimetic cultures and, especially, how secretome content influence on macrophage activity for therapy. hLMSCs from human donors were cultured on devices developed in house that enabled lung-mimetic strain. hLMSC secretome was analyzed for typical cytokines, chemokines and growth factors. RNA was analyzed for the gene expression of CTGF and CYR61. Human monocytes were differentiated to macrophages and assessed for their phagocytic capacity and for M1/M2 subtypes by the analysis of typical cell surface markers in the presence of hLMSC secretome. CTGF and CYR61 displayed a marked reduction when cultured in lung-derived hydrogels (L-Hydrogels). The secretome showed that lung-derived scaffolds had a distinct secretion while there was a large overlap between L-Hydrogel and the conventionally (2D) cultured samples. Additionally, secretome from L-Scaffold showed an HGF increase, while IL-6 and TNF-α decreased in lung-mimetic environments. Similarly, phagocytosis decreased in a lung-mimetic environment. L-Scaffold showed a decrease of M1 population while stretch upregulated M2b subpopulations. In summary, mechanical features of the lung ECM and stretch orchestrate anti-inflammatory and immunosuppressive outcomes of hLMSCs.

## 1. Introduction

Mesenchymal stromal cells (MSCs) have been demonstrated to possess immunoregulatory and anti-inflammatory functions. This, in combination with their ability to grow ex vivo, make them very attractive for cell therapy. MSC-based therapies for inflammatory lung diseases, such as acute respiratory distress syndrome (ARDS) and SARS-CoV-2 infection (COVID-19), rely mainly on the paracrine actions of MSCs [1,2,3]. Notably, MSCs have been demonstrated to have the ability to reprogram macrophages. For example, the secretion of prostaglandin E2 (PGE2) and TNF-stimulated gene 6 (TSG6) promotes the differentiation of M1 macrophages (pro-inflammatory) into M2 macrophages (anti-inflammatory) [4,5]. Interestingly, recent studies have demonstrated that M2 macrophages consist of at least three different phenotypes with different functions: M2a, M2b and M2c [6]. However, there is less information as to what extent MSC secretome influences M2 macrophage sub-phenotype differentiation.

Despite extensive research and several clinical trials in relation to inflammatory diseases, the treatment efficiency of using MSCs with a focus on the lung is debated [7,8,9]. Acknowledging that there are most likely several factors responsible for the lack of efficiency, there remains a fundamental lack of knowledge as to the mechanoactivation of MSCs by the biophysical lung microenvironment. In their native tissue, MSCs are constantly influenced by multiple external stimuli, including the compositional and structural properties of the surrounding extracellular matrix (ECM) and mechanical loading from breathing. Increasing research has been directed towards optimizing pre-culture conditions in order to mimic the native environment and, thereby, improve the therapeutic outcome of MSCs. For example, we previously reported that biophysically preconditioned MSCs showed an enhanced repair capacity compared to non-preconditioned MSCs [10]. Moreover, utilizing ECM-derived materials that enable a physiomimetic culture of cells by providing mechanical cues similar to their in vivo microenvironment (which entails intracellular mechanosignaling processes) [11,12] may boost the immunoregulatory and adhesive properties of MSCs, properties that are key features of their therapeutic efficacy.

Dynamic biophysical stimuli, such as shear stress and cyclic stretch exerted during breathing, are important parts of the physiomimetic environment, shown to be important for maintaining tissue homeostasis [13]. These, in combination with material properties, affect cellular activity. The mechanosensing pathways are, essentially, activated through the establishment of focal adhesions to the substrate, which, in turn, activates intracellular signaling, such as talin and focal adhesion kinases (FAK), which transmit these cues as biochemical signaling via RhoA/ROCK activity [14]. These intracellular pathways converge into the master gene expression regulator YAP (and its homolog TAZ), where connective tissue growth factor (CTGF, also known as CCN2) and cysteine-rich angiogenic inducer 61 (CYR61, also known as CCN1) are well-known downstream effectors of YAP [15].

The aim of this study was to evaluate the therapeutically relevant functions of human lung-derived MSCs (hLMSCs) under the influence of physiomimetic lung ECM conditions subjected to stretch and to explore the phenotypic changes exerted on macrophages by the secretome of hLMSCs cultured under these different physiomimetic lung conditions.

## 2. Materials and Methods

Unless otherwise specified, reagents for cell culture were purchased from Thermo Fisher Scientific, Waltham, MA, USA, and the reagents employed for non-cellular protocols were purchased from Sigma-Aldrich, St Louis, MO, USA.

### 2.1. Isolation of hLMSC

Adult lung-derived MSCs (hLMSCs) were isolated from peripheral transbronchial (parenchymal tissue) biopsies from 6 different donors, as described elsewhere [16] (a descriptive table of the main characteristics of the patients is presented in Appendix A). Briefly, lung biopsies were cut into small pieces and washed, and cells were isolated by enzymatical digestion with a mix of 300 U/mL collagenase type I, 1 mg/mL hyaluronidase and DNAse in Dulbecco’s phosphate-buffered saline (DPBS). After digestion, cells were washed and seeded and allowed to adhere to standard plastic tissue flasks with StemMACS MSC Expansion medium (Miltenyi Biotec, Bergisch Gladbach, Germany) supplemented with 10% FBS (Hyclone Laboratories, Logan, UT, USA) and 1% antibiotic–antimycotic. Medium was changed weekly, and cells were passaged with TrypLE Express at 70–90% confluence. After passage 2, hLMSC medium formulation was changed to DMEM supplemented with FBS (10%) to expand the cells prior to the experiments. Cells were used between passage 4 and 7.

### 2.2. Preparation of the Lung Scaffolds

Lung scaffolds (L-Scaffold) were obtained from pig lungs; tissue cubes of ~1 × 1 × 1 cm^3^ were sliced to 350 µm and decellularized with a protocol described elsewhere [17]. Briefly, the slices were incubated in a decellularization solution containing 8 mM CHAPS, 1 M NaCl and 25 mM EDTA in PBS. The incubation was performed in motion with 1 mL/slice of decellularization solution, which was changed six times over a period of four hours at room temperature. Then, slices were submerged in benzonase working buffer (20 mM Tris–HCl, 2 mM Mg^2+^ and 20 mM NaCl at pH 8) and incubated for 30 min at 37 °C. Decellularized slices were rinsed in PBS and stored until use at +4 °C in PBS containing 1% antibiotics.

### 2.3. Preparation of the Lung-Derived Hydrogels

Lung-ECM-derived hydrogels (L-Hydrogels) were prepared from porcine lung tissue from 3 different animals obtained from a local slaughterhouse. Briefly, the lung tissue was first decellularized by using an adapted protocol from Pouliot et al. [18], where the lungs were perfused through the trachea and the vasculature with 0.1% Triton X-100, sodium deoxycholate, DNase and 1 M sodium chloride, with intermediate perfusion with distilled water and PBS for rinsing purposes. Afterwards, decellularized lungs were cut into small pieces and frozen at −80 °C, freeze dried (Telstar Lyoquest-55 Plus, Terrassa, Spain) and pulverized into micron-sized particles at −180 °C using a cryogenic mill (SPEX 6755, NJ, USA). The resulting powder was digested at a concentration of 20 mg/mL in a 0.01 M HCl solution with pepsin from porcine gastric mucosa (1:10 concentration) under magnetic stirring at room temperature for 16 h. Pregel solution was then pH-stabilized to 7.4 (±0.4) by using 0.1 M NaOH and added PBS 10X and then frozen at −80 °C for subsequent use. 

### 2.4. Stretch Device and Functionalization

A custom-made device consisting of 3 wells per unit was built based on a previous design described elsewhere [19]. Samples were mounted on an elastic polydimethylsiloxane (PDMS) membrane (Gel-Pak, Hayward, CA, USA) of 150 µm and deflected to obtain ~20% of circumferential stretch at a frequency of 0.2 Hz. In order to promote either cell adhesion or attachment of DLS and hydrogel, the PDMS membranes were functionalized with collagen I. Briefly, the membranes were plasma-irradiated, treated with 10% APTES, rinsed and treated with 5% genipin. After final rinses, membranes were dried overnight. Next day, membranes were UV-sterilized and incubated with a collagen I solution (0.3 mg/mL) and washed prior to either cell seeding or attachment of DLS and hydrogels. The DLS and hydrogels were placed onto the membrane and allowed to attach overnight [20].

### 2.5. Physiomimetic Culture of hLMSC

L-Scaffolds were repopulated with a concentrated suspension of hLMSCs (40,000 cells/cm^2^), incubated for 1 h prior to adding 2 mL of culture medium and then incubated for another hour under mild agitation (60 rpm). L-Hydrogels were formed by mixing the pre-gel solution with hLMSCs to obtain a mixture of 400,000 cells/cm^3^ and then poured gently onto the treated wells. Cell-laden L-Hydrogels were crosslinked at 37 °C in the incubator 1 h prior to adding 2 mL of medium. The plastic controls were seeded with 8000 cells/cm^2^. The seeded cells for all conditions were incubated for 24 h in DMEM containing 10% Fetal Bovine Serum (FBS) prior to starting the mechanical stimulation. The medium was changed for DMEM supplemented with 2% of FBS, and the stretch devices were connected to the mechanical stimulation for 96 h with daily changes of medium (DMEM with 2% FBS) for all conditions. During the final 24 h, the cells were cultured in DMEM without FBS. The experiment was repeated with 6 different healthy donors and was assayed in triplicate. At the endpoint, the conditioned medium from all groups was collected and centrifuged 1000× *g* for 5 min at 4 °C and stored at −80 °C. The repopulated DLS and hydrogels were either saved for histological evaluation or RNA analysis. 

### 2.6. Confocal Immunofluorescence

Repopulated L-Scaffolds and L-Hydrogels were fixed in 4% formaldehyde for 45 min at room temperature and stored in PBS with 0.05% sodium azide (Sigma-Aldrich) at 4 °C until further processing. This included permeabilization in 0.1% Triton X-100 (Sigma-Aldrich) for 5 min at room temperature and staining with 1X Phalloidin-iFluor 555 (Abcam, Cambridge, UK) for F-actin staining and DAPI (nuclei staining) prepared in 1% BSA in PBS for 1 h at room temperature. The L-Scaffolds and L-Hydrogels were immersed in Ce3D (prepared as previously described [21]) in chambers built on top of slides using iSpacers (SunJin Lab) and glass coverslips. They were imaged in a resonant scanner A1RHD confocal microscope (Nikon) controlled with the NIS Elements AR software (Nikon) using a 10× air objective (Nikon) and laser excitation. Images were corrected for brightness and prepared for publication in NIS Elements AR Analysis software (Nikon).

### 2.7. Multiplex Secretome Evaluation

Conditioned medium from endpoint was assayed for 13 markers by a custom-made, multiplexed ELISA plate (R&D, Minneapolis) that detected: IL-1ra, m-CSF, MCP-1, MIP-2, IL-1β, IL-6, IL-8 and TNF-α, IFN-γ IFN S γ-10, VEGF, IL-4 and HGF. The data were normalized to cell number, and UMAP plots were created in RStudio (2021.09.1 based on R 4.1.2) with the umap package (0.2.7.0) and ggplot2 (3.3.5). The other graphs were made using GraphPad Prism (9.3.1). The vinyl graphs were plotted as parts of a whole and normalized in between each group.

### 2.8. Phagocytosis Assay

Human monocytes (THP-1, TIB-202, ATCC) were pre-cultured according to the manufacturer’s protocol. A total of 750,000 cells/cm^2^ were seeded in a 96-well plate and differentiated into adherent macrophages in medium supplemented with 50 ng/mL of phorbol 12-myristate 13-acetate (PMA) over 48 h. Phagocytosis was measured as described by the Vybrant Phagocytosis assay kit. Briefly, the culture medium was removed, and an equal number of fluorescence particles were mixed with conditioned medium from all culture conditions and added to the wells. After 3 h of incubation, excess particles were removed and quenched within the well by trypan blue. The net intake of particles was quantified according to the fluorescence readout of the test samples and deducted the acellular controls. Data was expressedaccording to the control DMEM 0% unexposed to hLMSCs.

### 2.9. Phenotype Assessment of Macrophages Exposed to MSC Secretome

Differentiated macrophages (THP-1) were polarized to M1 and M2 phenotypes by adding medium containing either LPS (50 ng/mL) or IL-4 (20 ng/mL) and IL-10 (10 ng/mL) (R&D Systems, Minneapolis, USA). Conditioned medium from the hLMSC cultures was mixed at 1:1 ratio with the medium of the M1 and M2 macrophages. M0, M1 and M2 controls were cultured in respective medium. After 24 h of culture, cells were prepared for flow cytometry on a BD LSRFortessa X20 and analyzed using FlowJo (10.8.1). Samples were cleaned in FlowJo using the FlowAI and then gated on FMO controls. A complete gating strategy can be found in the Appendix A. The following antibodies were used: CD11c (BV650, 563403 BD), CD40 (PE-Cy7, 561215 BD), CD80 (BV711, 751726 BD), CD86 (R718, 751920 BD), CD124 (BB700, 745925 BD), CD163 (BV786, 741003 BD), CD200R (BV421, 566344 BD), CD206 (APC, 550889 BD), CD281 (PE, 12-9911-42 eBioscience), HLA-DR (BB515, 564516 BD), 7AAD (7AAD, 555816 BD), 7AAD (7AAD, A9400 Sigma). Two 7AAD antibodies were used due to availability, but separate FMO controls were used for gating. All antibody samples were diluted in Brilliant stain buffer, 50 µL (563794 BD).

### 2.10. qPCR for Mechanosensory Markers

Samples for RNA extraction were collected at the endpoint and later stored in RNA (Thermo Fisher Scientific, Massachusetts) until further use. RNA from physiomimetically cultured hLMSCs was extracted using the mirVana RNA isolation kit (Thermo Fisher Scientific, Massachusetts) and, later, total RNA was quantified. RNA from samples cultured directly onto the bare plastic was isolated using the RNeasy Mini Kit (Thermo Fisher Scientific, Massachusetts). Afterwards, all samples were retro-transcribed using the Quantitect Reverse Transcription Kit. Changes in gene expression of the connective tissue growth factor (CTGF) and the cysteine-rich angiogenic inducer 61 (CYR61) were quantified by quantitative PCR using the QuantiFast SYBR Green Master Mix in a StepOne Plus thermocycler. Relative CTGF and CYR61 was normalized to the expression of peptidylprolyl isomerase A (PPIA) and to the expression of the plastic ST group by the 2−ΔΔCt method [22]. All reagents for nucleic acid protocols were purchased from Qiagen unless otherwise specified.

### 2.11. Statistical Analysis

Data were presented as individual values for each output where the bars represent the mean. Differences among groups were analyzed by two-way repeated measurements analysis of variances (ANOVA) for the parameters of environment (Plastic, L-Scaffold and L-Hydrogel) and strain (static-ST or stretch-CS). Specific variation between conditions were assayed by Sidak’s multiple comparison test. The specific differences within the environments were assessed by non-parametric paired *t*-test (Wilcoxon). All statistical analyses were performed with GraphPad Software (San Diego, CA, USA), considering *p* < 0.05 as statistically significant.

## 3. Result

### 3.1. Lung-Derived Biomaterials Harness Differential Mechanical Microenvironments

Primary hLMSCs derived from peripheral lung tissue from six healthy donors (four male/two female, age range between 43 and 81 years) were cultured in two different lung-mimetic conditions under either cyclic stretch (CS) or static conditions (ST) and compared to cells cultured on standard tissue culture plastic. As illustrated in Figure 1A, the mechanical confinement was different between the L-Scaffold and the L-Hydrogel, which was further demonstrated through 3D imaging (Figure 1B). hLMSCs in L-Hydrogels showed that cells distributed throughout the whole construct, while cells in L-Scaffold were mainly distributed on top of the ECM with partial infiltration along the stack and with more cell-to-cell contact. Regarding plastic conditions, cells displayed elongated spindle-like shapes on the surface suggesting some cell-to-cell interactions (Appendix A). No difference was seen between CS and ST conditions. RNA analysis for the well-known mechanical markers CTGF and CYR61 showed that LMSC cultured within L-Scaffold had a marked decrease in gene expression (five-fold) compared to L-Scaffold that had similar levels to plastic. Interestingly, the statistical analysis by two-way ANOVA showed significance for the lung-mimetic environments (*p* = 0.0468 for CYR61 and *p* = 0.0142 for CTGF) but not for the strain applied. Nevertheless, the specific, multiple comparisons between conditions showed no significant variations.

### 3.2. Secretome Content Is Influenced by the Microenvironment Due to the Physiomimetic Culture

To compare the secretome of hLMSCs cultured under different conditions, a multiplex ELISA was performed on the culture medium. The data plotted in UMAP graphs showed that secretome from the same donor clustered (Figure 2A). Most striking was the cluster for the secretome profile of L-Scaffold, while plastic and L-Hydrogel clusters showed some overlapping (Figure 2B). Comparing cyclic stretch to static conditions in the different environments, HGF (black) was the dominating factor in all conditions (Figure 2E). Furthermore, the HGF showed the highest values due to the L-Scaffold environments: 1915.18 and 1099.26 pg/mL normalized to the cell amount for static and cyclic stretch, respectively (Appendix A). Whereas L-Hydrogel environment had a lesser impact on HGF compared to L-Scaffold and plastic, the combination of L-Hydrogel and cyclic stretch elicited HGF increase. Other factors, such as MCP-1 and m-CSF, displayed a higher portion in Figure 2E, which was slightly decreased due to cyclic stretch. On a donor level, paired samples for cyclic and static stretch were analyzed (Figure 2F). Interestingly, IL-6 and TNF-alpha decreased in the medium from hLMSC cultured in L-Hydrogel and L-Scaffold under cyclic stretch conditions, while the level increased in cultures on Plastic under cyclic stretch.

### 3.3. Secretome from Physiomimetic Environments Attenuates Phagocytosis and the Stretch Increases M2b Subpopulations

To evaluate the functional capacities of the secretome from hLMSCs, conditioned medium from the different culture conditions was used to expose polarized macrophages recognized as M0, M1 and M2 subtypes. After exposure, the phagocytosis capacity of M0 macrophages was assessed (Figure 3A), as well as the phenotype expression of typical markers for M1, M2a, M2b and M2c (Figure 3B). The net intake of fluorescent bioparticles showed a decrease by the conditioning media from the L-Scaffold and L-Hydrogel under stretch (approximately to half), while exposure of the medium from hLMSCs cultured on plastic under stretch showed the opposite trend in phagocytosis with a two-fold increase (Figure 3A). The statistical analysis by paired test revealed a significant decrease due to the stretch in L-Hydrogel (*p* = 0.0317). To further evaluate the effect on the macrophage phenotype, the cells were stained for two typical membrane markers for M1 and M2 and, additionally, the M2a–c subtypes (Figure 3B). Figure 3C shows that the physiomimetic culture elicited slight changes in the M1 and M2 subpopulation regardless of environment or stretch applied. Remarkably, the percentages of macrophages expressing M1 markers were reduced by ~30% when exposed to the secretome from stretched L-Scaffold compared to the plastic static, while sustaining a similar M2 phenotype. Regarding the M2 subtypes (Figure 3D), the stretched environments elicited increases of 50%, 70% and two-fold (for plastic, L-Hydrogel and L-Scaffold, respectively) only for the M2b subpopulations.

## 4. Discussion

Increasing evidence points out that the microenvironment affects the paracrine signaling of MSCs and that different cues may be of different importance for the intended therapeutic outcome, motivating the optimization of the pre-culture conditions using bioengineering. Here, we demonstrated that the main differences in the secretome profiles of hLMSCs are intrinsic to the biomaterial nature, and the effect of dynamic mechanical stimuli is dependent on the mechanical confinement of the cells.

Despite the pulmonary origin of L-Scaffolds and L-Hydrogels, the structural properties are different in terms of stiffness, foremost in the ultrastructure, thus, fundamentally impacting cell organization and orientation. In L-Hydrogel, the cells are fully encapsulated (Figure 1B) within a highly hydrophilic meshwork, which is more similar to the native situation where the hLMSCs are interspersed in the interstitial matrix. On the other hand, the cell-to-cell interactions of hLMSCs found in the cultures in L-Scaffold have been reported to be key for mechanical sensation due to the physical constraint along the cell sheet [23]. CTGF and CYR61 are downstream effectors of YAP which play pivotal roles in integrating different mechanical cues and further induce a physiological reparative process in the organ; accordingly, we wanted to investigate their gene expression changes in our physiomimetic systems. Compared to plastic cultures, there was a striking decrease in gene expression of both CTGF and CYR61 in L-Hydrogel, indicating a diminished activity of YAP. Interestingly, this decrease was only seen for CYR61 in the L-Scaffold, which indicates that there is an interplay between the ultrastructure and mechanical confinement. The cell-to-cell interactions seem to primarily drive the response of CTGF, while CYR61 seems to be more closely linked to stiffness of the substrate as the difference in stiffness was neglectable between L-Hydrogel and L-Scaffold in comparison to plastic (KPa to GPa; Figure 1A) [24,25]. Adding cyclic stretch further indicated the role of the microenvironment. Cyclic stretch seemed to induce an overall reduction in gene expression in most donor cells cultured in the physiomimetic environments, while the opposite effect was observed in plastic. Apart from cell-to-cell interaction, the hLMSCs may sense strain patterns differently within the biomimetic materials due to their differential capacity of bearing deformation, specifically, re-orientation of the skewed components of the scaffold and the water motions in and out the hydrogel after each deformation [26].

Static and dynamic mechanical signals are integrated by intracellular YAP signaling. Interestingly, some studies have highlighted a link between anti-inflammatory actions of MSC and YAP signaling [27]. Analysis of hLMSC secretome pointed out different clusters due to donor variation, which several other studies raised as an important factor that may impact the therapeutic outcome in MSC treatments [28,29]. For this reason, it is of high relevance to develop culture conditions that can predict the “quality” of the intended population of MSCs for therapy. We and others have demonstrated that major changes can be induced in the MSC secretome and, therefore, its paracrine actions when cultured under physiomimetic cues [2,25,30]. Intriguingly, L-Hydrogel and plastic conditions bore some overlapping features in their secretome cluster profiles despite sizeable differences in CTGF and CYR61 expression (Figure 1C), while L-Scaffold displayed a distinct phenotype despite having a similar gene expression to plastic with regard to mechanoreceptors. YAP and their upstream Hippo pathway effectors showed reciprocal interactions with NFκB upstream effectors, contributing to the overall inflammatory response [27]. These findings could explain the divergent expression of CTGF/CYR61 and the secretome exhibited in the lung-mimetic environments.

From the vinyl representations in Figure 2E, where data were normalized to the total measured secretome and, therefore, indicate the influence of each marker on the secretome profile, it can be seen that different conditions had a fundamental impact on the cellular response of hLMSCs. Interestingly, HGF was found to be decreased within the L-Hydrogel environment compared to in L-Scaffold and plastic and was increased by the contribution of stretch. Previous studies attributed the reparative and immunomodulatory roles of MSCs in spinal cord injury models [31] and lung diseases to the actions of secreted HGF. Specifically, Kennelly et al. [32] showed in murine elastase-induced models of COPD that HGF played a central role in the cytoprotective, anti-apoptotic and tissue reparative effects of hMSCs, reverting the injurious consequences associated with the disease. Additionally, a lower proportion of HGFs from hLMSCs cultured in L-Hydrogel was accompanied by an increased proportion of the leukocyte-related chemokines MCP-1 and m-CSF, which may have a larger impact on circulating monocytes. While MCP-1 induces a chemotactic activity in circulating monocytes, m-CSF stimulates growth and differentiation of monocytes into macrophages [33,34]. IL-6 and TNF-alpha secretion, depicted in Figure 2F, showed an increased production of both cytokines by stretch on plastic environment and the opposite trend for lung-mimetic environments. Specifically, Saldaña et al. [35] showed that TNF-alpha from primed macrophages modulated the response of MSCs towards an anti-inflammatory phenotype, as depicted in their reduction in TNF-alpha. Concomitantly, neutralization of TNF-alpha reduced the aforementioned anti-inflammatory response. The decreased TNF-alpha in our studies indicated that cyclic stretch triggers an anti-inflammatory secretome from hLMSCs cultured in lung-mimetic environments and the opposite for the plastic conventional cultures. Similarly, IL-6 decreased under cyclic stretch in physiomimetic environments, which is in line with the switch towards the anti-inflammatory milieu. Taken together, L-Hydrogel, which displayed low values of HGF, low IL-6 and TNF-alpha, suggests a “quiescent” state concerning the immunosuppressive actions of hLMSCs, yet the higher values of MCP-1 and m-CSF indicate a possible relevant role during the acute phases of an inflammatory disease where recruitment and differentiation is key to counteracting an inflammatory insult. Furthermore, L-Scaffold subjected to cyclic stretch induced high values of HGF and lower IL-6 and TNF-alpha, suggesting that the secretome from this culture could trigger an immunosuppressive milieu to resolve precedent acute phases of inflammation.

To assess the pre-conditioning effect of the hLMSCs on the secretome, two functional assays with macrophages were carried out. Non-polarized macrophages exposed to secretome from cells cultured under cyclic stretch in lung-mimetic conditions revealed a seemingly clear difference with a decreased phagocytosis compared to the plastic condition where phagocytosis increased, showing similar trends in the secretion of L-6 and TNF-alpha (Figure 2E), suggesting that physiomimetic environments prompt MSCs towards a pro- or anti-inflammatory response. The influence on macrophage phagocytic activity by MSCs is communicated by macrophage-to-MSC interaction, and the inflammatory trigger drives the ultimate phagocytic outcome. Whereas direct transfer of mitochondria from MSC to macrophages increases phagocytosis and M1 functions in a microbial model of ARDS [36], Adutler-Lieber et al. [37] showed that MSCs decrease phagocytosis in a paracrine manner accompanied with M2 polarization. Additionally, this corroborates the hypothetic resolving actions of hLMSCs in an inflammatory disease by L-Scaffold plus cyclic stretch in a functional assay of secretome on polarized macrophages. Exposure of the medium from the physiomimetic conditions under cyclic stretch seemed to have an increased attenuating effect on M1 phenotype overall, which is in line with studies that assayed bone and hair regeneration [38,39]. Mechanically stimulated skin from mice, which contained hair stem cells and hydroxyapatite scaffolds repopulated with MSCs, triggered both an upregulation of M2 accompanied by a reduction in M1 phenotypes. Intriguingly, stretched L-Scaffolds showed the highest percentage of M1 subtype and, therefore, correlated with the anti-inflammatory effects of its secretome, as described above. Owing to the slight changes seen in M2 from the physiomimetic culture, we pursued analyzing specific M2 subtypes a, b and c. To our surprise, and regardless of the microenvironment, stretch was found to exclusively increase the M2b subtype. Other studies found that MSCs promote immunosuppression by inducing an increase of the M2b phenotype [4,40]. Specifically, Phillip et al. [41] showed that IL-1β and IFN-γprimed MSCs promoted an increase of M2b through the actions of PGE2. Accordingly, our setting of stretch behaved similarly and, thus, induced MSCs to elicit their immunosuppressive actions.

Taken together, L-Scaffold subjected to cyclic stretch displayed features of secretome that were positively relevant for the preconditioning of hLMSCs with the goal of treating inflammatory diseases. Furthermore, secretome elicited key changes in macrophage activity in favor of therapy and, thus, reinforced the notion of preconditioning cells under physiomimetic features of their mechanical surroundings. On the other hand, L-Hydrogel influenced hLMSCs towards a more quiescent status, which could be explained by the hydrophilic meshwork of immature fibers on display and the cell–ECM crosstalk. Finally, the concise actions of stretch within the culture were shown to be microenvironment dependent, highlighting the importance of dynamic stimuli in the culture of MSCs for therapy. Taking into account the wide spectra of questions that remain open, a major limitation in our study was the large variability introduced by the individual donor, where we saw different levels of cellular responses and an opposing effect of the mechanical loading. Another critical point is the origin from which MSCs were isolated. In this study, we used MSCs isolated from lung tissue; however, comparison of MSCs obtained from different origins, such as bone marrow, umbilical cord and adipose tissue, needs to be further studied. Additionally, to the best of our knowledge, this is the first study investigating the macrophage behavioral changes of M2 subtypes caused by the secretome of physiomimetically cultured hLMSCs. However, it also raises some concerns, specifically regarding the exact mechanisms behind secretome influencing M1 and the different M2s. Furthermore, what are the actions of increased M2b due to cyclic stretch or, in more general terms, how can we positively modulate macrophage phenotypes in therapy? Moreover, these were all in vitro studies, and additional testing of the immunomodulatory effects on MSCs grown in our model system needs to be carried out in proper in vivo experiments.

## 5. Conclusions

In summary, this study presents a culture platform incorporating both physiomimetic lung ECM conditions and mechanical stimulation of MSCs that foster their anti-inflammatory and immunosuppressive paracrine actions with the translational possibility to achieve genuine MSC-based therapeutic approaches. Additionally, it opens a broad avenue of mechanistic studies to bridge the link between mechano-related and inflammatory pathways of MSCs.

## Figures and Tables

**Figure 1 cells-11-01866-f001:**
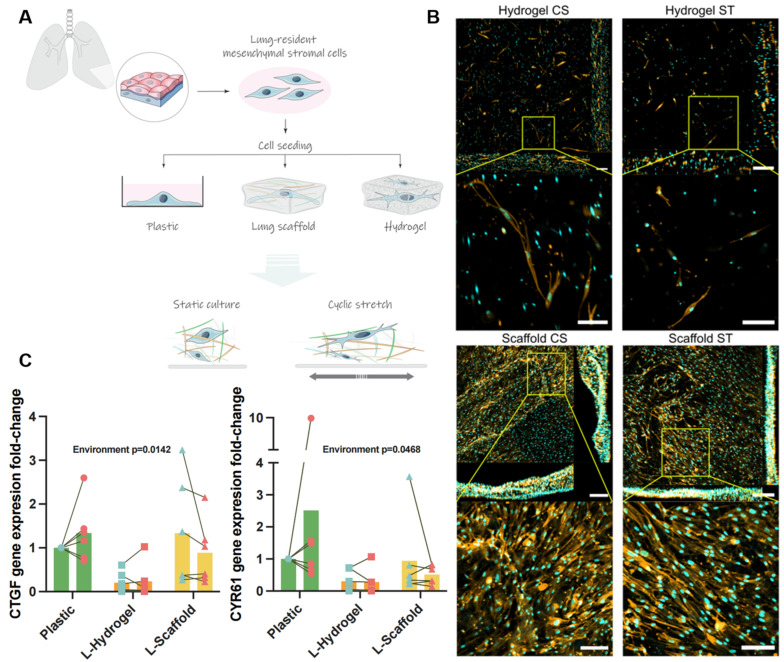
(**A**) Workflow to achieve physiomimetic culture of hLMSCs with mechanical cues from the lung; (**B**) 3D images of the physiomimetic culture of hLMSCs. Left side of the image is a 2D representation of the stack captured. In addition, a close-up image is attached showing the cell morphology and dispersion within the biomaterial. (**C**) Gene expression of CTGF and CYR61 expressed in fold changes compared to the plastic ST group.

**Figure 2 cells-11-01866-f002:**
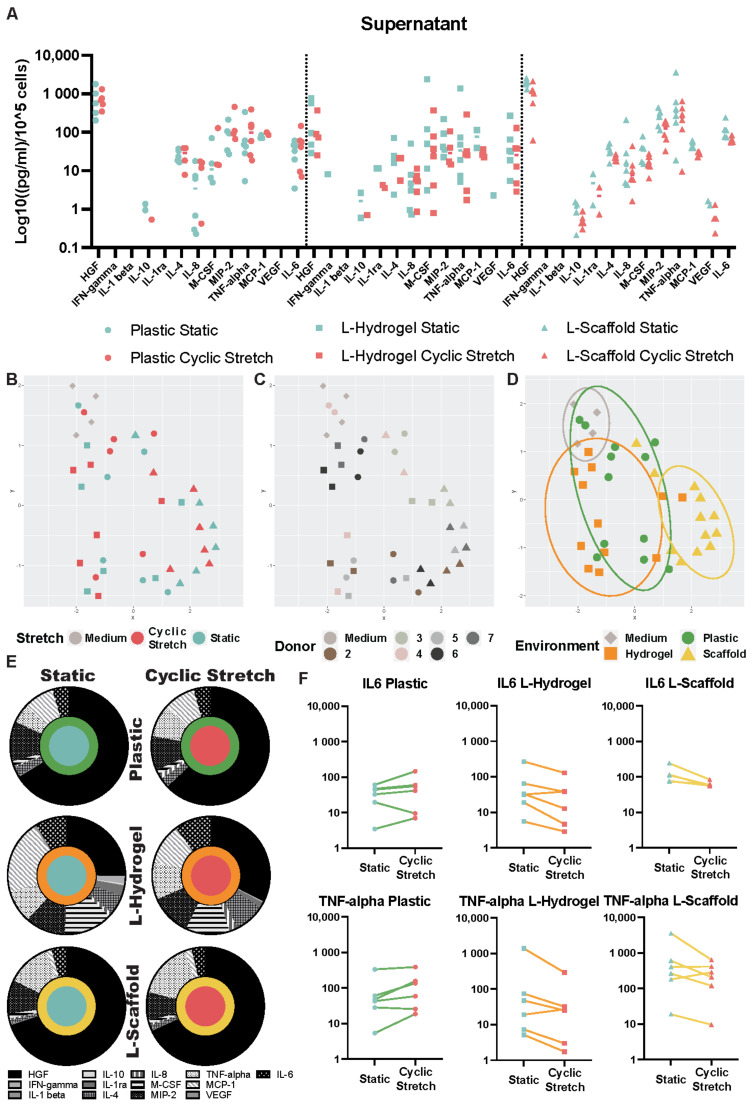
Secretome profiling of hLMSCs under lung-mimetic signals. (**A**) Secreted soluble mediators normalized to cell amount of the culture. (**B**–**D**) UMAPs representation of hLMSC secretion pointing out proximity of the samples among donors and the microenvironment. (**E**) Vinyl graphs showed influence of each marker on the whole secretion assayed in the multiplex ELISA. (**F**) IL6 and TNF-alpha secretion, highlighting the specific effect of CS for each donor.

**Figure 3 cells-11-01866-f003:**
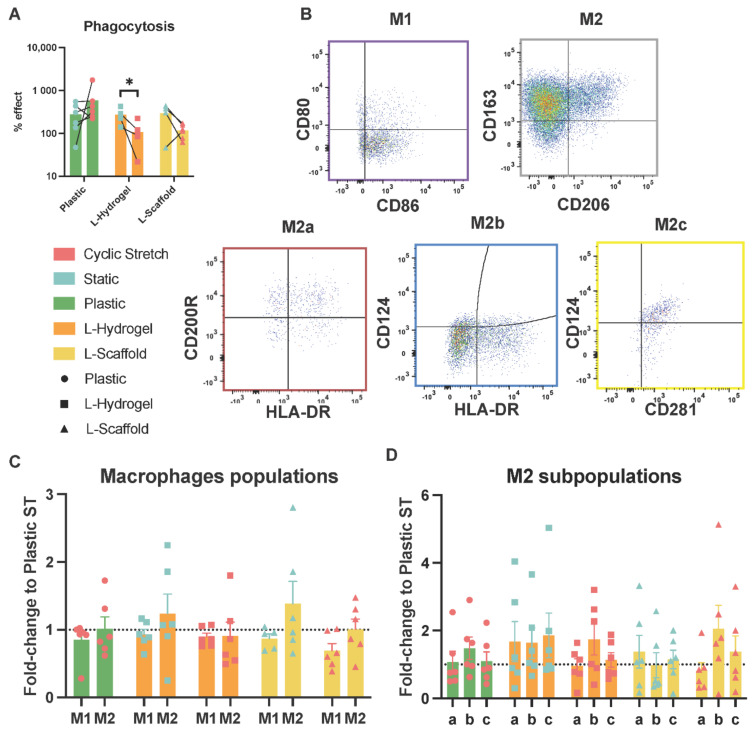
Macrophage activity modulated by the secretome of physiomimetically cultured hLMSCs. (**A**) Phagocytosis is expressed as % effect compared to the positive control of the assay. * *p* = 0.0317. (**B**) Gating strategy to sort the macrophage population and the M2 subpopulations; (**C**,**D**) Macrophage subpopulations expressed in fold changes to plastic ST. Bars show mean values for the different conditions assayed: green (plastic), orange (L-Hydrogel) and yellow (L-Scaffold).

## Data Availability

Data supporting the findings of this study are available from the corresponding author upon reasonable request.

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
