# Peer review of "hLMSC Secretome Affects Macrophage Activity Differentially Depending on Lung-Mimetic Environments"

_cells, 2022, doi:10.3390/cells11121866_

Round 1
Reviewer 1 Report
Falcones et al have shown that human lung mesenchymal stromal cells (MSC) modulate macrophage phenotype functions differentially depending on the lung mimetic environments. The study was well designed and he experiments were meticulously conducted. They have grown the MSC on engineered lung-mimetic matrices with or without mechanical stress to study their influence on MSC growth and their secretome which modulates the macrophage phenotype functions. However, there are few changes are reuired to accept in its present form.
Major
- In Figure 1A, authors shematically illustrated the growth kinetics of MSC on L-scafolds and L-hydrogels but provided the images of cell morphology only L-hydrogels cultures. Since it is very important phenotype, it required to show the images of all cultyre condition including plastic.
- Fig 1C needs to be clearly labelled. Although authors claim the 2-way ANOVA was performed for statistical analysis, it is not shown in Figure C and in fugure legend. IT is also reuired to compare between the culture conditions.
Minor
- Page 1, line 40, It is stated that 'M2 macrophages consist 4 different phenotypes' , it should be 3 different phenotypes.
- Page2, line 67, CYR6 need to be changed to CYR61.
Author Response
Dear reviewer 1,
We are grateful for the insightful and thorough review. Please find attached a point-by-point response. With these modifications, we believe that our revised manuscript will meet your criteria for publication.
Best regards,
Bryan Falcones.

Reviewer 2 Report
In the current manuscript, authors describe the effect of their in-house physiomimetic culture platforms on the secretome content of hLMSCs. They show that lung-derived scaffolds resulted in a distinct secretion while there was a large overlap between the L-hydrogel and the conventionally (2D) cultured samples. They specifically report differences in HGF, IL-6 and TNF-α levels after exposure to different physiomimetic culture platforms. Additionally, they report the effect of different culture platforms on the anti-inflammatory and immunosuppressive outcomes of hLMSC function and their effect on macrophage phenotype and function.
Overall, the authors report interesting findings related to the effect of their in-house physiomimetic culture platforms on the secretome content of human lung-derived MSCs, however, translating the proposed pre-conditioning approach to clinical application of hLMSCs may not seem to be feasible, specifically if MSCs are isolated from the lung of individuals during an infection.
Considering the fact that bone marrow derived, umbilical cord-derived of fat-derived MSCs are frequently being applied in clinical trials, the same procedure should have been tested on one of those cell types as well.
Additionally, testing the immunomodulatory effects of the current primed MSCs in animal models of inflammation or infection is highly recommended. Indeed, it is necessary to study the macrophages polarization and their anti-microbial effects in vivo considering the fact that many pre-conditioning approaches were reported to work perfectly in vitro, and not in vivo.
Author Response
Dear reviewer 2,
We are grateful for the insightful and thorough review. Please find attached a point-by-point response. With these modifications, we believe that our revised manuscript will meet your criteria for publication.
Best regards,
Bryan Falcones.

Round 2
Reviewer 2 Report
Authors did not revise the manuscript substantially. By adding more to the discussion, the quality of this work was not improved/. Authors need to conduct the actual experiments suggested in the original comments.
Author Response
We agree with the reviewer about the relevance of the experiments proposed in our system, however we are not currently able to do these experiments due to the lack of ethical permits and of time to perform them. Accordingly, that is why we adjusted the scope of the manuscript to a short communication instead. Nevertheless, we acknowledge the ideas and thus are strongly considering performing these experiments in future projects.